# Information Encoded by the Flavivirus Genomes beyond the Nucleotide Sequence

**DOI:** 10.3390/ijms22073738

**Published:** 2021-04-03

**Authors:** Sara Ramos-Lorente, Cristina Romero-López, Alfredo Berzal-Herranz

**Affiliations:** Instituto de Parasitología y Biomedicina López-Neyra (IPBLN-CSIC), Av. Conocimiento 17, Armilla, 18016 Granada, Spain; seramos@correo.ugr.es

**Keywords:** *Flavivirus*, RNA genomes, interactome, structure/function of RNA domains, long-distant RNA–RNA interactions, sfRNAs

## Abstract

The genus *Flavivirus* comprises numerous, small, single positive-stranded RNA viruses, many of which are important human pathogens. To store all the information required for their successful propagation, flaviviruses use discrete structural genomic RNA elements to code for functional information by the establishment of dynamic networks of long-range RNA–RNA interactions that promote specific folding. These structural elements behave as true cis-acting, non-coding RNAs (ncRNAs) and have essential regulatory roles in the viral cycle. These include the control of the formation of subgenomic RNAs, known as sfRNAs, via the prevention of the complete degradation of the RNA genome. These sfRNAs are important in ensuring viral fitness. This work summarizes our current knowledge of the functions performed by the genome conformations and the role of RNA–RNA interactions in these functions. It also reviews the role of RNA structure in the production of sfRNAs across the genus *Flavivirus*, and their existence in related viruses.

## 1. Introduction

The genus *Flavivirus* comprises many small, enveloped viruses commonly known as flaviviruses. Of global health importance, they are responsible for many emerging and re-emerging outbreaks of human disease, with dengue virus (DENV), zika virus (ZIKV), Japanese encephalitis virus (JEV), yellow fever virus (YFV) and West Nile virus (WNV) among the most important. Flaviviruses can be largely classified according to their arthropod vector as either mosquito-borne flaviviruses (MBFV), tick-borne flaviviruses (TBFV), no-known vector flaviviruses (NKFV), insect-specific flaviviruses (ISFV), or as the unclassified Tamana bat viruses (TABV) (Table 1) [1,2].

The flavivirus genome consists of a single positive-stranded RNA molecule with just one open reading frame (ORF) flanked by untranslated 5′ and 3′ regions (UTRs). The ORF encodes a polyprotein that is processed to produce three structural and seven non-structural viral proteins [3,4]. The RNA genome is endowed with a type I cap structure at the 5′ terminus and lacks a poly A tail at its 3′ end. Like many other RNA viruses, flaviviruses have developed different strategies to expand their capacity to store information, allowing them to code for all the information required for the successful completion of the viral cycle in a genome of just ~11,000 nucleotides. Besides containing protein-coding information, the flavivirus RNA genome stores essential information in structurally conserved units. These are scattered throughout the genome, with the flanking UTRs being rich in them (reviewed by [5,6]). Thus, RNA genomes behave as molecules that store protein-coding information (mRNA), and as non-coding RNAs (ncRNAs), functions that together ensure the correct regulation of the viral cycle. This multifunctional behavior is maintained by the establishment of intricate networks of RNA–RNA interactions—the RNA interactome—that allow the formation of high-order structures critical to infection.

This review examines our current knowledge of the structure/function of the flavivirus genome, with special attention given to the networks of interactions that occur among different structural RNA units. An overview of the involvement of genomic RNA structural elements in the production of subgenomic flavivirus RNAs (sfRNAs) across the genus *Flavivirus*, and their putative existence in related viruses, is also provided.

## 2. Structural Diversification of the Flavivirus RNA Genome

All flaviviruses bear UTRs at both ends of their genome; these are essential for viral viability. Their functioning relies on their structure, which varies widely among members of different ecological groups and even different clades of the same group. UTR folding involves a number of discrete, functionally active structural RNA units with defined roles. It is noteworthy that, regardless of their different folding, overall UTR functioning is preserved across the flaviviruses, revealing that this functionality can be achieved with different RNA conformations.

### 2.1. 5′ Untranslated Region

The 5′UTR is ~100 nt long [7]. Although its sequence varies somewhat across flaviviruses, the overall RNA structure is largely conserved (with some more notable differences between members of the MBFV and TBFV groups). The 5′UTR folds into two conserved domains (Figure 1): stem-loop A (SLA), which operates as a promoter of viral replication [8,9,10], and stem-loop B (SLB), which contains the essential 5′UAR (upstream AUG region) motif required for genome cyclization (see below) [11,12]. Both domains are linked by a short U-rich sequence, which can pair with the viral translation starting region to generate the 5′UAR-flanking stem (UFS) (Figure 1) [13]. The UFS is present in most MBFV members, in some representatives of the NKFV, and in different members of the ISFV group [13], while the YFV clade and certain members of NKFV show a different folding pattern resulting in a structure termed ψUFS (Figure 1) [13]. The UFS and ψUFS structures of the MBFV and NKFV members contribute to the recruitment of the viral polymerase NS5 while the viral genome is in linear form [13,14]. In the ISFV, its function remains unknown. Intriguingly, this structure is absent in the TBFV; rather, a short hairpin with a large apical loop (5′SL2) is located in an analogous position (Figure 1b) [13]. This suggests that flaviviruses may have developed alternative mechanisms for the control of RNA synthesis [13].

### 2.2. 3′ Untranslated Region

The 3′UTR region is between 400 and 800 nt long. A remarkable feature of this region is the existence of structural duplications. The existence of alternative folding intermediates dependent on duplicated elements increases the functional diversity. Indeed, such functional diversification is known to exist, providing, for example, adaptation to different hosts.

The 3′UTR is organized into three major domains in the MBFV (Figure 2) [15]:

(i) Domain I, located downstream from the translation stop codon, contains a variable structural element followed by a number of conserved stem-loops (SLs). Indeed, significant variations in the number of SLs have been reported. For example, some members of the MBFV group, such as YFV, possess tandem duplications of two SLs (known as RYF) (Figure 2a) [16]. While DENV shows a duplicated SL—SL-I and SL-II (Figure 2a)—other representative MBFV members, such as those belonging to the JEV clade, have duplicated this cassette and thus have four SL units (SL-I to SL-IV) (Figure 2a) [17,18,19].

The existence of SL duplications in domain I of different flaviviruses may be an adaptive mechanism that assists in the movement between natural hosts without interfering with any other activities of the 3′UTR [20]. For example, in DENV, point mutations in SL-II that reduce its structural stability led to a loss of replication efficacy in mammalian cells [21], but not in mosquito cells [22].

An important structural feature of flavivirus RNA genomes is that they bear a variable number of conserved pseudoknot structures (PK) in domain I (Figure 2). They involve the residues located at the apical loop of each SL and the nearby downstream sequences. These SL-PK elements influence infectivity, host adaptation and viral fitness [21,23,24,25], and are likely related to the stalling of cellular exoribonuclease XRN1, leading to the production of sfRNAs [26] (see below). Importantly, when a single SL-PK structural element is borne, as in YFV (PK1; Figure 2a), virulence correlates with the stability of that SL-PK [27].

(ii) Domain II shows moderate variations in its structural organization, not only between different ecological groups but also among different clades of the same group [28]. For example, in members of the JEV and DENV clades within the MBFV group, domain II shows two dumbbell (DB)-like structures, the 5′ and 3′DBs [29], with a linker sequence varying in sequence and length (Figure 2a) [28]. In contrast, YFV bears a single pseudo-DB element, ψDB, the sequence of which seems to be a combination of the duplicated 5′ and 3′DB elements present in other MBFV members (Figure 2a) [17,20,24]. In addition, it bears a complete DB structural element. Representatives of NKFV, such as MODV (Table 1), have a single DB element (Figure 2c), while in ISFV and TBFV, this kind of structure is substituted by moderately conserved SLs (Figure 2b,d) [24,30]. Interestingly, duplicated elements commonly undergo functional specialization, as can be inferred from the sequence analysis of the DB elements in DENV. In this virus, the homology between the 5′ and 3′DB is less than that observed between DB elements of different viral serotypes [20], a consequence of the different selective pressure placed on each DB element in adult mosquitoes. Only the 3′DB element shows sequence variation (it can be even lost), while the 5′DB element is critical to the preservation of efficient replication. These observations support the idea that DB duplications can widen the range of genome functionality, something that might occur in all members of the MBFV group [20]. In MBFV, the DB elements also play a role in translation [31], although their involvement in the generation of sfRNAs and replication are their most significant contributions to the viral cycle [31,32]. Again, duplicated elements provide for different structure-based viral adaptations [21].

A significant feature of flavivirus genomes with double DB elements is the formation of at least one PK structure stabilized by four-way junctions (Figure 2). The PK elements within domain II involve sequences located at the apical loops (named TLs) of the DB and the corresponding downstream complementary motifs. Since sequences interacting with the TLs overlap conserved sequences (CS) required for genomic cyclization (Figure 2) [19,24,33], it is likely that these PK structures compete for elements that are essential in genomic cyclization, preventing the latter from occurring (see below; [20,25,34]). This competition may provide a means of regulating switching between different steps of the viral cycle.

The DB elements also incorporate conserved and reverse-conserved sequences, named CS2 and RCS2, respectively (Figure 2) [29,35]. These sequences of 20 to 45 nt are mostly conserved in flaviviruses [29]. CS2 appears in the DENV, JEV and YFV clades, as well as in the NKFV group, while RCS2 is absent from YFV and NKFV (Figure 2). Duplications of conserved sequences may have evolved to maximize protein recruitment and thus enhance replication, translation, etc., but this remains to be confirmed.

(iii) Domain III is the most structurally conserved region within the 3′UTR. It contains a large terminal SL-3′SL, preceded by a small hairpin structure, sHP (Figure 2). Both 3′SL and sHP are preserved across MBFV, TBFV and NKFV [30,36], while sHP is absent in cISFV [37].

The sHP element is a small hairpin carrying different concatenate sequence motifs (UAR, DAR (downstream AUG region) and CS) and is involved in genomic cyclization. It can adopt different conformations in the cyclized and linear forms [38]. Systematic mutation methods used with DENV have revealed the preservation of the hairpin folding to be essential for the replication of the virus in mammalian cells, though tolerance to nucleotide mutations is relatively high. However, sequence variations can interfere significantly with replication in mosquito cells, suggesting a double function for sHP, both at the sequence level and the structural level [22]. The strong conservation of sHP across all ecological groups confirms this element to be essential to flaviviruses fitness.

3′SL bears a conserved pentanucleotide sequence motif (5′-CACAG-3′) in the apical loop [37,39,40] that is required for replication [41,42,43,44] and the potentiation of translation [45,46]. These functions are achieved via the recruitment of different proteins including viral NS5 polymerase [47], eukaryotic elongation factor eEF-1α [48,49,50], T-cell intracellular antigen-1 (TIA-1) and TIA-1-related (TIAR) proteins [51,52,53], Y box-binding protein-1 (YB-1; [54]), interleukin enhancer binding factor 3 (ILF3, also referred as nuclear factor 90 [NF90] [55]), DEAH box polypeptide 9 (DHX9, also known as RNA helicase A [RHA] [55]), and La protein [56,57].

Two conserved base pairs can be also found at the base of the 3′SL element in the JEV and DENV clades, which are flanked by a metastable structure identified as two symmetrical bulges (Figure 2) [58,59]. This region stabilizes the whole 3′SL structure, and it has been proposed to be involved in the switch from the linear to the cyclized genome [59,60]. Indeed, the 5′ basal segment of the 3′SL contains the 3′UAR motif, which is essential for genome cyclization (Figure 2) ([44,60]; see below). The unwinding of this segment is required for the efficient replication of MBFV in mammalian cells, while in mosquito cells, such structural destabilization occurs only marginally (see below; [59]).

It would thus appear that a dynamic network of RNA–RNA interactions regulates the critical processes of the viral life cycle and the switches between them. The diversification of RNA genome functionality, including successful replication in two ecologically different hosts, is determined by the acquisition of new structural RNA elements.

## 3. Cyclization of the Flavivirus Genome

From a structural point of view, one of the most interesting and complex flavivirus phenomena is the acquisition of a circular closed-loop topology by the RNA genome (the cyclization process) [38]. The acquisition of this conformation was revealed in 1987 [29], but electron microscopy had already recorded the same behavior in alphaviruses during the 1970s [61]. Many unrelated viruses have since been shown to do the same [62,63,64,65,66,67]. In flaviviruses, genome cyclization is necessary for replication [6,11,23,68,69]; it ensures the synthesis of full-length genomes [29]. It also provides a control mechanism for translation [31,70] and infectivity [71] and helps to govern transitions between the different steps of the viral cycle [72].

In flaviviruses, cyclization depends on the establishment of non-covalent interactions between complementary sequences at the 5′ and the 3′ ends of the genome [73]. In MBFV, three pairs of complementary sequence motifs are responsible:

(i) The cyclization sequence 5′CS [29,39,74] located at the 5′ end of the ORF, which pairs with the complementary 3′CS motif (also named CS1) at the base of the 3′SL (Figure 1, Figure 2 and Figure 3). The complementarity between these sequence motifs has been demonstrated by different authors [11,23,74,75] who also showed that the preservation of base-pairing, independent of the actual nucleotides present, was enough to ensure efficient viral replication. The length of this sequence was found to be 10, 11 and 18 nt for the members of the DENV, JEV and YFV clades, respectively [11,76]. In the TBFV group, two pairs of cyclization sequences—5′-3′CS-A and 5′-3′CS-B—have been recorded (Figure 3). The CS-B pair is located in a position analogous to the CS motifs in MBFV, whereas the CS-A pair is displaced to the ends of the genome: thus, the 5′CS-A sequence lies upstream of the AUG codon and the 3′CS-A is located at the base of 3′SL (Figure 3) [39,74]. Both pairs differ in sequence from the CS motifs described in MBFV [77].

(ii) The 5′ downstream sequence of the AUG region, 5′DAR, and its partner 3′DAR, which overlaps with the sHP element (and therefore the 3′CS motif) in the linear conformation of viral RNA (Figure 2 and Figure 3) [69,78,79]. The formation of the DAR pair is essential for virus replication, but mismatches or low complementarity between the 5′DAR and the 3′DAR are better tolerated in mammalian than in mosquito cells [69]. Thus, the DAR interaction is subject to differential selection pressure and contributes to host adaptation [21,69]. Interestingly, with the exception of the clades YFV, SPOV and DENV, the remaining MBFV members contain two pairs of DAR sequences, DARI and DARII (Figure 3) [28]. In addition, while 3′DAR motifs seem to be conserved across flaviviruses, the 5′DAR sequences are only conserved across DENV serotypes [78].

(iii) The 5′ upstream sequence of the AUG region, 5′UAR, and its complementary sequence 3′UAR within the basal portion of the stem in the 3′SL element (Figure 3) [11,12]. The formation of the so-called UAR pair requires the unwinding of the UFS at the 5′UTR, which is likely mediated by cellular factors such as AUF-1 [58]. Destabilization at the base of 3′SL within the 3′UTR is also needed (Figure 1 and Figure 2) [12,13,59,80].

The compilation of functional data [20,80,81,82,83] has led to a cyclization model being proposed in which the initial interaction involving the CS and DAR motifs induces the unwinding of the sHP at the base of the 3′SL, thus releasing the 3′UAR sequence to form the UAR pair. The 5′UAR-3′UAR interaction would then stabilize the tertiary structure and promote high-order structural changes [72]. In TBFV, genome cyclization is further stabilized by an additional kissing-loop interaction between the conserved hexanucleotide motif of the 5′SL6 element and its complementary sequence at the 3′UTR (3′SL3) (Figure 1, Figure 2 and Figure 3) [84].

The cyclization process can be facilitated by additional cis-structural elements. For example, different MBFVs form a three-stemmed PK downstream of the 5′CS (DCS-PK) (Figure 1 and Figure 2) [85], which, together with the UAR, DAR and CS pairs, generates a functional domain that regulates the conformational switch inducing the cyclization of the genome. Interestingly, this region is highly conserved in MBFVs, with the exception of the YFV clade, in which a single hairpin has substituted the DCS-PK (Figure 1). Whether this hairpin performs an equivalent function to that shown by DCS-PK remains to be seen.

Genome cyclization may fit into the intracellular viral infection process as follows: in its linear form, the actively translated RNA genome presents the UFS duplex, which is crucial for NS5 recruitment to the SLA element. This induces a translationally repressed state [13]. Genome cyclization then occurs via the establishment of long-distance RNA–RNA interactions, and the recruitment of viral proteins and cellular factors [56,58,81,86,87,88]. The UFS then unwinds. This conformational effect promotes the transfer of the NS5 to the free 3′ end, initiating the replication of the circular form [9,44,89]. The UFS may therefore be considered a switch for controlling the transition from viral translation to replication. It has also been proposed that the UFS acts as a structurally dynamic element to control NS5 recruitment during replication in a manner dependent on viral RNA levels [90], but this needs to be confirmed.

## 4. Subgenomic Flavivirus RNAs

In addition to the functional roles of the discrete 3′UTR elements in the formation of the secondary and tertiary structures described above, the 3′UTR participates in the production of small, flavivirus-specific, non-coding RNAs known as sfRNAs. sfRNAs result from the incomplete degradation of the viral RNA genome; they accumulate in the cytoplasm of infected cells [91].

### 4.1. Biogenesis of sfRNAs

Uncapped viral genomic RNAs are digested by the 5′-3′ exoribonuclease XRN1 in mammalian cells, or by its homolog Pacman in insect cells [26,92]. This digestion ends, however, if the enzyme encounters specific 3D structures in the 3′UTR known as XRN1-resistant RNA structures (xrRNAs), thus producing sfRNAs. The amounts of different sfRNAs produced, and the function of these different types, varies between flaviviruses. Host alternation during the viral cycle also determines the production of different sfRNA species [93,94].

The members of the MBFV group show differences in their production of sfRNAs. Members of the JEV clade bear the SL-II element, which participates in the formation of xrRNA1 (Figure 4); when XRN1 encounters the latter, digestion stops and sfRNA1 is produced (Figure 4) [26]. Other sfRNAs are produced as a result of additional xrRNA elements downstream of SL-II, indicating that the presence of duplicated elements leads to the duplication of the tertiary ‘stop’ structures. The appearance of shorter sfRNAs also indicates that XRN1 is not completely blocked by the xrRNA1 site [15]. Indeed, three additional species of sfRNAs (sfRNA2, sfRNA3 and sfRNA4) have been described in WNV, the result of XRN1 stalling at xrRNA structures involving the SL-IV (xrRNA2), 5′DB (xrRNA3) and 3′DB (xrRNA4) structural elements, respectively (Figure 4a) [26,32,95]. In contrast, only two sfRNAs (sfRNA1 and 2), have been identified in human-adapted DENV, the result of XRN1 stalling at SL-II- and SL-IV-dependent xrRNA sites, respectively (Figure 4a) [94,96]. A similar pattern has been seen in ZIKV-infected mosquito and human cells [97], although a third sfRNA (200 nt-long) species might also be produced in mosquitoes [98]. Unlike that seen for the JEV and DENV clades, the members of YFV produce two sfRNAs via the influence of a single xrRNA involving the SL-E structural element (Figure 4a). Both of these sfRNAs share the 5′ end but vary in the 3′ end, as sfRNA2 lacks the 3′SL; the mechanism behind its generation is unknown, as is its function [93,99]. So far, such truncated sfRNAs have only been observed in YFV [100]. Two sfRNAs have been reported in TBEV (Figure 4b), and a single sfRNA has been detected for the ISFV and NKFV group members (Figure 4c,d) [101,102,103].

Alternative pathways leading to the accumulation of sfRNAs have been proposed. One suggests the existence of promoter sequences close to the XRN1 stalling site, as seen in the 3′UTR of JEV clade members. In this case, subgenomic RNA production would derive from RNA transcription. This assumption is further supported by the observation that, in these viruses, sfRNAs accumulate even more strongly in XRN1-knock down cells [104]. A similar strategy might be followed by distantly related plant viruses of the family *Tombusviridae* [105], suggesting a convergence in the evolution of subgenomic RNA production systems among different viruses.

### 4.2. 3D xrRNA Folding Is Preserved across Flaviviruses

The overall folding of xrRNAs is well-conserved across flaviviruses, although significant variability in the primary and secondary structural elements can be found. Different sequence motifs and secondary structures can therefore induce the same three-dimensional conformation.

Several experimental strategies have been used to elucidate the architecture of the flavivirus 3′UTR xrRNA structures, the secondary structure elements, and the tertiary interactions involved in their formation. Although all xrRNAs share a common three-dimensional structure [103,106], differences in the secondary structure [93,94] have led to two major classes of xrRNA structures being defined: (i) class 1, which comprises xrRNAs present in the 3′UTR of MBFV and ISFV members; and (ii) class 2, which includes those xrRNAs found in TBFV and NKFV members [101]. Some authors have contested this division, arguing that the tertiary interactions of the class 2 xrRNAs structures can be used to predict those of class 1 [103]. However, the results of a recent study combining bioinformatics, biochemical and structural analysis suggest the further redefinition of class 1 into two subclasses: 1a and 1b [107]. Although full consensus is yet to be reached, most authors agree that the use of the two-class classification system aids for a more comprehensive understanding of xrRNAs and provides a tool for searching for new xrRNAs in different viral systems.

All xrRNA structures found across the genus *Flavivirus* share three SLs that fold into a three-way junction [32,95,96,100,106]. These SLs are connected by RNA–RNA interactions to form a PK. Depending on the class of xrRNA, these interactions range from canonical Watson–Crick contacts to electrostatic stabilizations. In all cases, the xrRNA folds into a ring-like structure in which the 5′ end of the RNA passes through it, forming a kind of knot. The helicase activity of XRN1 is unable to unfold this ring-like structure, leading to the blockage of viral genome degradation and the formation of sfRNAs [93,97,100,106]. *In vitro* assays have shown that the xrRNA ring-like structure also blocks 5′-3′ viral RNA degradation performed by unrelated exoribonucleases [101]. However, it does not impede the advance of enzymes that work in the 3′-5′ direction, such as viral NS5 polymerase [100,108]. Thus, xrRNA structures mechanically block XRN1 via the formation of a specific three-dimensional structure rather than establishing specific interactions with any enzyme, or by promoting the thermodynamic stabilization of the xrRNA [101].

A clear example of how different primary and secondary structures can lead to the same 3D xrRNA has been reported in the divergent flavivirus Tamana bat virus (TABV) [30,107,109,110]. X-ray crystallization studies have shown that while the ring-like structure of the xrRNA is preserved across TABV and other flaviviruses, significant variations occur in the corresponding nucleotide sequences and secondary structural elements [107,110]. One major difference resides in the shortening of the stems that form the ring-like structure. In order to accommodate the required tertiary interactions associated with these shorter helices, the overall structure is bent, a feature not seen in MBFV xrRNAs with longer stems. Consequently, in TABV, the xrRNA is more compact and rigid. In addition, the region encircling the 5′ end of the xrRNA lacks Watson–Crick pairs, requiring the interplay of Mg^2+^ ions and water molecules to stabilize the final structure [110]. These differences support the idea that the TABV represents a phylogenetically distinct lineage within flaviviruses and mark the path for understanding the mechanisms that drive the generation of subgenomic RNAs in other virus families (see below). The characterization of the TABV’s xrRNAs provides additional structural information for searching new stalling sites in other RNA viruses.

### 4.3. The Role of sfRNAs

The requirement of XRN1 for the biogenesis of sfRNAs involves the shortage of this exoribonuclease for physiological cellular mRNA turnover, promoting cytopathic effects [26,111,112]. Cytopathicity also occurs through the direct inhibition of the Bcl-2 protein by sfRNAs.

sfRNAs are also involved in strategies that interfere with antiviral responses. One of these strategies involves the recognition and binding of isoforms of the protein Dicer along with protein components of the interference machinery. Thus, proteins essential to the cell for the physiological production of miRNAs are sequestered. This also prevents the generation of siRNAs from double-stranded regions of the flavivirus genome [113]. Flaviviruses can also evade the type I interferon-mediated response. This involves the use of different viral proteins, as described for other members of the family *Flaviviridae*. It also occurs via the direct sequestration by sfRNAs of cellular factors—such as TRIM-25, G3BP1/2 and CAPRIN1—that participate in the interferon response pathway [114]. Protein recruitment may require specific sequence motifs within the 3′UTR, but given the variation shown by this region, other unknown mechanisms likely help impair the interferon-mediated response.

The importance of sfRNAs as driving elements in the evolution of flaviviruses has been reviewed by Slonchak et al. (2018). The accumulation of mutations in specific domains within the 3′UTR modifies the genomic:subgenomic RNA ratio; larger numbers of sfRNA may help to promote outbreaks of disease [100]. It should also be remembered that duplicated secondary structural elements within the 3′UTR account for specific sfRNA production patterns in different hosts, helping them to switch between different natural hosts [24,93].

### 4.4. xrRNAs in Other Viruses

Since XRN1 can completely degrade the RNA of any viral RNA genome, the existence of xrRNAs and sfRNA-like subgenomic RNAs in viruses other than flaviviruses was thought to be unlikely [26]. However, the discovery of xrRNAs in TABV suggested that they might actually exist [30,107,115,116,117,118,119,120]. In fact, the structural divergences observed in the xrRNA of TBAV are also seen in members of other genera of *Flaviviridae* family. Computational studies have identified putative xrRNAs in representative members of the genera *Pestivirus*, *Pegivirus* and *Hepacivirus*, expanding the presence of these structural elements to the entire *Flaviviridae* family [107] and supporting the definition of the structural subclass 1b [107].

The simple presence of xrRNAs does not necessarily dictate the production of subgenomic RNAs. For example, viruses with subclass 1b xrRNAs do not produce them (and interestingly they do not infect different hosts), unlike the viruses containing subclass 1a xrRNAs that alternate between two host. In many cases, these 1b xrRNAs do not locate to the 3′UTR of the viral genome [107,115,119]. It may be that they function as translation and/or replication switches, rather than being involved in the production of functional subgenomic RNAs, but more information is required to determine exactly what they do.

The phenomenon of XRN1-stalling mediated by xrRNAs extends beyond the family *Flaviviridae*. Following their search for subgenomic RNAs, Charley et al. [117] reported that phleboviruses and arenaviruses (which belong to different families within the order *Bunyavirales*) contain structural elements in their 3′UTR that can efficiently stall XRN1 leading to the production of sfRNA-like subgenomic RNAs. Interestingly, these elements differ in their primary and secondary structure to those described for members of the family *Flaviviridae*. For example, XRN1 stalling in phlebovirus seems to be dependent on G-rich stretches, similar to the poly(G) sequences reported to stall XRN1 in yeast. Nevertheless, additional conformational determinants must be necessary to block XRN1 activity since not all G-rich sequence motifs promote this effect [117]. In the case of arenaviruses, multiple subgenomic RNAs are produced from the different 3′UTRs contained in their genomes, via the incomplete degradation mediated by XRN1 [117]. Surprisingly, these UTRs do not show G-rich sequence stretches, nor any other predicted sequence motif required for xrRNA formation. Therefore, different viruses may adopt different structural strategies for controlling RNA turnover, many of which are probably still to be discovered.

Finally, up to 40 putative xrRNAs have been reported in different members of the families *Tombusviridae* and *Luteoviridae* [105,116,119,120], all of which infect plants. It is remarkable that many of these xrRNAs locate to intergenic regions rather than just 3′UTRs, generating heterogeneous pools of subgenomic RNAs. This suggests an additional role for xrRNAs in the control of viral protein synthesis [119]. Greater structural variability has also been reported for this group of xrRNAs. Their optimal folding depends on the presence of additional sequences and secondary structure elements. This implies a loss of modularity as well as an increase in the final size of the structural element in certain members of the family *Tombusviridae* [105]. In contrast, in other plant viruses such as BNYVV (beet necrotic yellow vein virus from the family *Benyviridae*) or RCNMV (red clover necrotic mosaic virus, belonging to the family *Tombusviridae*), the minimum core for generating the xrRNA is smaller than that described for flaviviruses [116,120].

Taken together, these observations again highlight that ring-like folding relies on a plethora of different secondary structural elements that together govern the stalling of XRN1.

## 5. Conclusions and Perspectives

The conservation of the overall structure of flaviviral RNA genomes ensures their correct functioning and passage through the viral cycle. However, this structure can depend on the equilibrium between different foldings governed by a dynamic network of regulated RNA–RNA interactions (the interactome). Flaviviruses have evolved through the acquisition and preservation of discrete structural RNA units distributed throughout the genome; the RNA–RNA interactions among them determine genome functionality. The switch between essential functions (e.g., translation, replication) is finely regulated and governed by changes in the above network of interactions. This leads to the acquisition of structures competent for the recruitment of the viral and/or cellular factors needed for a specific function. Different flaviviruses may show different structural solutions for the same function. Knowledge of the essential roles played by viral RNA genome structures may provide new targets for combating RNA virus infections.

## Figures and Tables

**Figure 1 ijms-22-03738-f001:**
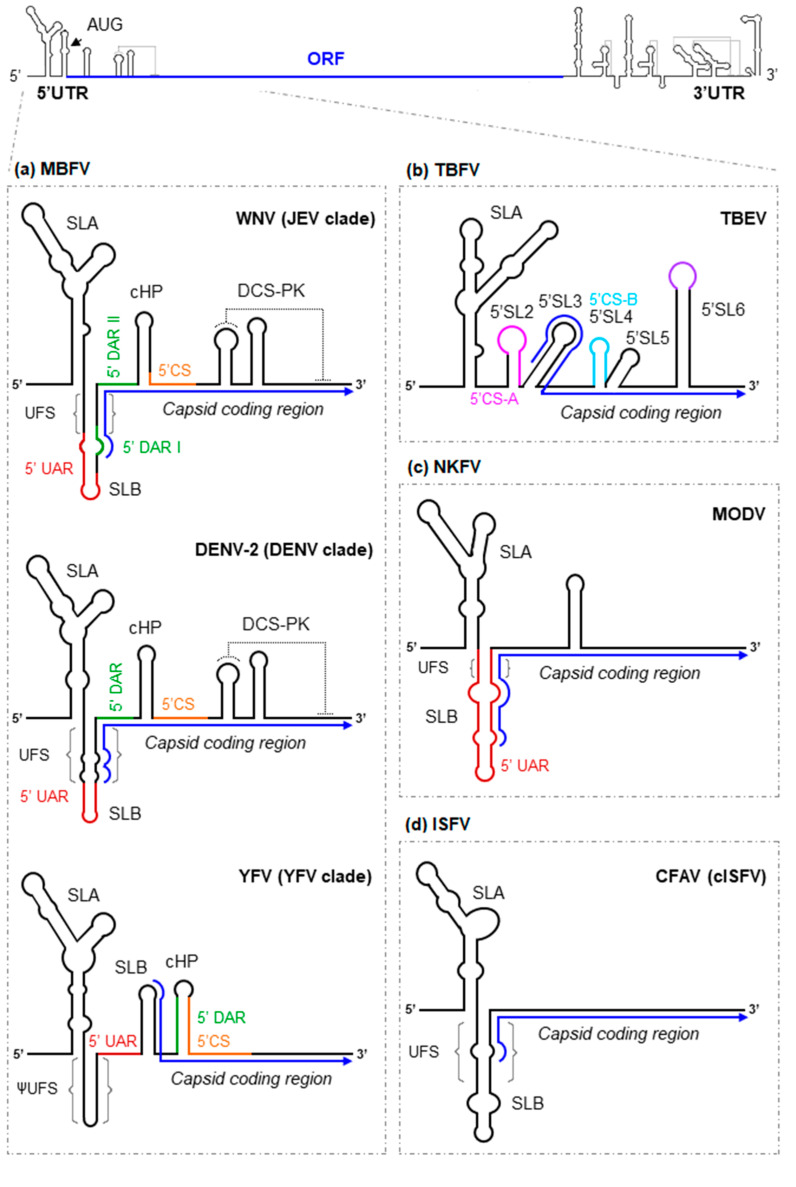
Structural RNA elements of the 5′ end of representative flavivirus genomes. As a model for flaviviruses, a diagram of the WNV genome is shown at the top of the figure. The arrowhead indicates the position of the AUG translation initiation codon. The ORF is represented by a thick blue line. Enlarged are diagrams representing the main functional–structural RNA elements and sequence motifs identified at the 5′ end of the genome of representative members of the: (**a**) mosquito-borne flaviviruses group (represented by WNV, DENV-2 and YFV); (**b**) tick-borne flaviviruses group (represented by TBEV); (**c**) No-known vector flaviviruses group (represented by MODV); and (**d**) insect-specific flaviviruses group (represented by CFAV). The blue arrow line indicates the ORF. Structural elements and sequence motifs are named. Sequence motifs involved in genome cyclization (UAR, DAR and CS) are indicated by colored lines. Dotted lines indicate interactions involved in the formation of pseudoknot structures.

**Figure 2 ijms-22-03738-f002:**
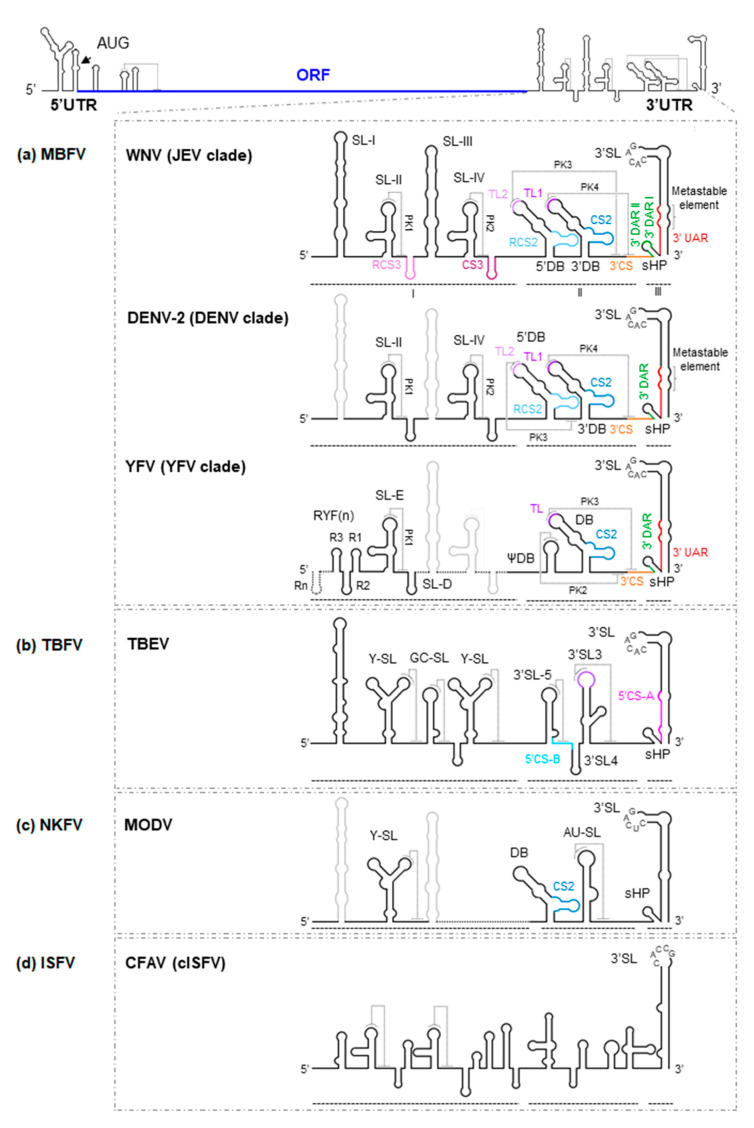
Secondary structure model of the 3′UTR of flavivirus genomes. As a model for flaviviruses, a diagram of the WNV genome is shown at the top of the figure. Below are detailed representations of the secondary structure of the genomic 3′UTR, indicating the identified secondary structural elements, the tertiary interactions and sequence motifs, for (**a**) WNV, DENV-2 and YFV, all members of the MBFV group; (**b**) TBEV, representative of the TBFV group; (**c**) MODV, a member of the NKFV group; and (**d**) CFAV, a member of the ISFV group. Secondary structural elements of each virus are indicated (thick black lines) on the secondary structure model of the WNV background (thick grey lines); note the differences between them. Structural elements and sequence motifs are named. Sequence motifs involved in genome cyclization (UAR, DAR and CS) are indicated by colored lines (same key as in Figure 1) to designate corresponding sequence partners in both figures. Dotted lines indicate interactions involved in the formation of pseudoknot structures. Thick dotted lines in the representation of YFV secondary structure indicate the presence of extra RYF elements in other members of the YFV clade.

**Figure 3 ijms-22-03738-f003:**
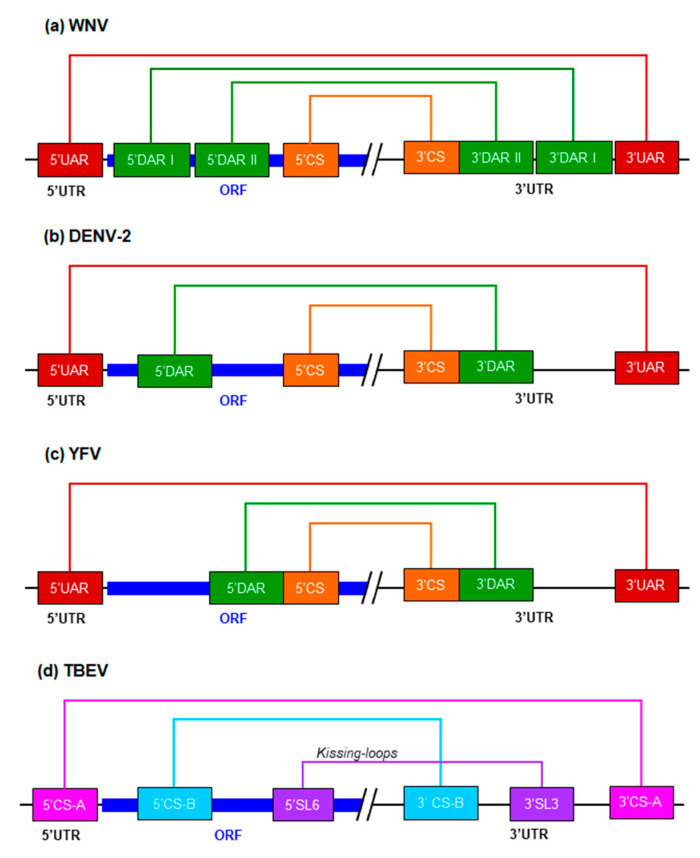
Long-range RNA–RNA interactions involved in the cyclization of flavivirus genomes. The diagram shows the linear form of the RNA genomes of four representative flaviviruses: (**a**) WNV; (**b**) DENV-2; (**c**) YFV; and (**d**) TBEV. Colored boxes (same key as in Figure 1 and Figure 2) show different interacting sequences with their names. Long-range interactions are illustrated by thick colored lines. The dark blue box represents the ORF.

**Figure 4 ijms-22-03738-f004:**
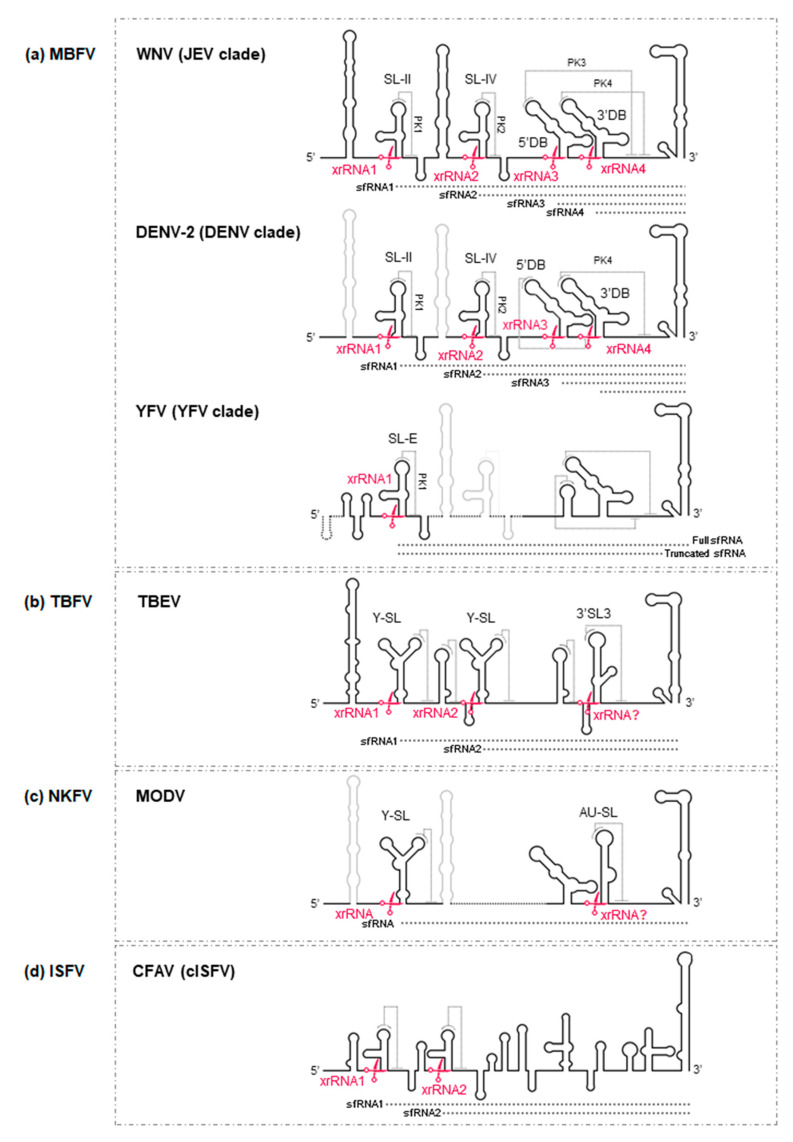
sfRNAs pattern. Diagram of the proposed xrRNA within the 3′UTR of the RNA genome of representative flaviviruses of: (**a**) the MFV; (**b**) TBFV; (**c**) NKFV and (**d**) ISFV groups. xrRNA elements are indicated by red scissors. xrRNA? denotes that the corresponding putative sfRNA has not been identified. sfRNAs are represented by dotted black lines. The key for the lines is as described for Figure 2.

**Table 1 ijms-22-03738-t001:** Classification of flaviviruses.

Name	Abbreviation
**Mosquito-borne** **flaviviruses ***	**MBFV**
Dengue virus clade	DENV
Dengue virus serotype 1	DENV-1
Dengue virus serotype 1	DENV-2
Dengue virus serotype 1	DENV-3
Dengue virus serotype 1	DENV-4
Japanese encephalitis virus clade	JEV clade
Japanese encephalitis virus	JEV
West Nile virus	WNV
Spondweni virus clade	SPOV
Zika virus	ZIKV
Yellow fever virus clade	YFV clade
Yellow fever virus	YFV
**Tick-borne flaviviruses ***	**TBFV**
Mammalian tick-borne virus	
Tick-borne encephalitis virus	TBEV
**No-known-vector flaviviruses ***	**NKFV**
Modoc virus	MODV
**Insect-specific flaviviruses ***	**ISFV**
Classic insect-specific flavivirus clade	cISFV
Cell fusion agent virus	CFAV
**Divergent Flaviviruses ***	
Tamana bat virus	TABV

* Only those groups mentioned in the manuscript.

## Data Availability

Not applicable.

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
