# Peer review of "Information Encoded by the Flavivirus Genomes beyond the Nucleotide Sequence"

_ijms, 2021, doi:10.3390/ijms22073738_

Round 1
Reviewer 1 Report
The review manuscript entitled “Information encoded by the flavivirus genomes beyond the nucleotide sequence” by Berzal-Herranz and co-workers describes the structural elements in the flavivirus RNA genome and their functional significance in the viral cycle. The review discussed diversity in viral genome RNA secondary and tertiary structures at their 3’ UTR and 5’UTR and explained the role of these structures in different subdomains. They also defined how long-range RNA: RNA interaction at their 3’ UTR and 5’UTR helps in the cyclization of the flavivirus RNA genome. Along with this, they also mentioned the biogenesis of sub-genomic flavivirus RNAs (sfRNAs) by incomplete degradation of the viral RNA genome and their functional significance in viral fitness. In brief, this review elucidates the function of cis-acting factors including the structure and sequence of flavivirus RNA genomes in passage through the viral cycle as well as viral fitness.
I think this is a good analysis of Flavivirus genomic RNA structures and their functional significance and it should be published in the present form.
Author Response
We thank reviewer 1 for the positive evaluation. Minor changes in grammar and spelling have been made in order to follow the reviewer suggestions.
Reviewer 2 Report
Ramos-Lorente et al. prepared an article about the structural features of the flaviviruses. They outlined the classification according to the vector and analyzed the best-known representatives of each of them. The central part of the article is about the structural diversity of 5' and 3' regions, RNA-RNA interactions that enable cyclization of the genome, and the role of xrRNAs.
The article presents a complete review of the current state of knowledge regarding flaviviruses' structure. It has helpful, well-designed diagrams, and the primary matter of the text is written with great care. Each abbreviation is explained, and every topic has a short introduction, which improves the reading experience and comprehension. The bibliography choice is also excellent, as it covers both the classical articles on the subject and the latest discoveries.
I find the knot-like tertiary structure of the xrRNAs especially intriguing. The article might benefit from adding its visualization or a diagram. Still, I understand there exist already other articles focused solely on xrRNAs, so I leave this only as a mild suggestion to the authors.
At line 157 (Section 2.2, part about Domain III), I have found one occurrence of a leftover citation in another format: (D. J. Gritsun et al., 2014).
Author Response
We thank the reviewer for the positive evaluation. Minor changes in grammar and spelling have been made in order to follow the reviewer suggestions. The citation included in line 157 has also been formatted according to the journal style.
We also thank the reviewer for the suggestion of including a figure showing the intricate structure of xrRNAs. We agree that it would be really helpful. However, including a 3D folding image would require holding the copyright permission. On the other hand, we feel that picturing the secondary structure of the xrRNA for a specific flaviviral clade could not be illustrative enough. In order to balance this lack of information, we have provided a complete set of references in which the xrRNA folding is clearly illustrated.